# Theoretical Determination of Size Effects in Zeolite-Catalyzed Alcohol Dehydration

**Larissa Y. Kunz** [1,†], **Lintao Bu** [1,†], **Brandon C. Knott** [1], **Cong Liu** [2], **Mark R. Nimlos** [1], **Rajeev S. Assary** [3], **Larry A. Curtiss** [3], **David J. Robichaud** [1] and **Seonah Kim** [1,*] 

[1]   National Renewable Energy Laboratory, 15013 Denver West Pkwy, Golden, CO 80401, USA
[2]   Chemical Sciences and Engineering Division, Argonne National Laboratory, 9700 South Cass Avenue, Lemont, IL 60439, USA
[3]   Materials Science Division, Argonne National Laboratory, 9700 South Cass Avenue, Lemont, IL 60439, USA
*   Correspondence: Seonah.Kim@nrel.gov; Tel.: +1-303-384-7323
†   These authors contributed equally to this work.

**Abstract:** In the upgrading of biomass pyrolysis vapors to hydrocarbons, dehydration accomplishes a primary objective of removing oxygen, and acidic zeolites represent promising catalysts for the dehydration reaction. Here, we utilized density functional theory calculations to estimate adsorption energetics and intrinsic kinetics of alcohol dehydration over H-ZSM-5, H-BEA, and H-AEL zeolites. The ONIOM (our Own N-layered Integrated molecular Orbital and molecular Mechanics) calculations of adsorption energies were observed to be inconsistent when benchmarked against QM (Quantum Mechanical)/Hartree–Fock and periodic boundary condition calculations. However, reaction coordinate calculations of adsorbed species and transition states were consistent across all levels considered. Comparison of ethanol, isopropanol (IPA), and tert-amyl alcohol (TAA) over these three zeolites allowed for a detailed examination of how confinement impacts on reaction mechanisms and kinetics. The TAA, seen to proceed via a carbocationic mechanism, was found to have the lowest activation barrier, followed by IPA and then ethanol, both of which dehydrate via a concerted mechanism. Barriers in H-BEA were consistently found to be lower than in H-ZSM-5 and H-AEL, attributed to late transition states and either elevated strain or inaccurately estimating long-range electrostatic interactions in H-AEL, respectively. Molecular dynamics simulations revealed that the diffusivity of these three alcohols in H-ZSM-5 were significantly overestimated by Knudsen diffusion, which will complicate experimental efforts to develop a kinetic model for catalytic fast pyrolysis.

**Keywords:** biomass pyrolysis; alcohol dehydration; zeolite; DFT; ONIOM

## 1. Introduction

With its high carbon and hydrogen contents, bio-oil derived from biomass pyrolysis presents a viable alternative to petroleum as a nearly carbon-neutral precursor to liquid fuels with a high energy density relative to raw biomass [1]. However, the high oxygenate content of bio-oil renders it immiscible with conventional non-polar fuels and leads to fuel instability and corrosion issues [2–4]. These oxygenates come in a variety of forms, including aldehydes, ketones, carboxylic acids, phenolics, and alcohols.

A fraction of the oxygen present in the pyrolysis vapors can be removed via de-oxygenation reactions that extract oxygen primarily in the forms of water, carbon monoxide, and carbon dioxide over zeolite catalysts (either in situ, contemporaneous with pyrolysis, or ex situ, in a downstream upgrading reactor) [5]. For example, rice husk pyrolysis studies over an H-ZSM-5 catalyst have demonstrated that water is formed predominantly at lower temperatures (400 to 500 °C), whereas

higher temperatures (500 to 600 °C) favor the formation of CO and $CO_2$ [6,7]. The dehydration reactions at lower temperatures are preferable because they remove oxygen without removing carbon, and carbon yield is highly correlated with fuel selling price [8]. Such dehydration reactions are likely to be rate-limiting for a variety of reaction classes, including Diels–Alder cycloaddition and carbon–carbon coupling reactions of carbonyls [9,10]. Hence, an ability to control dehydration kinetics within zeolite pores can be desirable.

To be able to tune the dehydration selectivity of a potential catalyst, the dehydration mechanisms for the various oxygenates must first be understood. These mechanisms and, hence, the product distributions are dependent on structural and electronic properties of the system; relevant structural properties include the zeolite pore and cavity size and shape, which determine the steric interactions between the substrate and active site. Jae et al. [11], for instance, observed a relatively high coke yield and low oxygenate yield when using catalysts with large pores (e.g., SSZ-55 zeolite, Beta zeolite, Y zeolite) for the catalytic fast pyrolysis of glucose compared to small or medium pore-sized zeolites. Theoretical calculations can help to elucidate reaction mechanisms proposed for oxygenate–zeolite pairings to explain such product distributions.

Alcohols are the simplest model substrates allowing for investigation of dehydration reaction barriers and associated reaction pathways. Greenhalf et al. [12] demonstrated that alcohols can account for ~5–7% of pyrolysis oil using straw, perennial grasses, and hardwoods as feedstock. Biomass-derived alcohols can undergo dehydration reactions to produce light olefins [13,14], which, in turn, can be converted into aromatic hydrocarbons over zeolite catalysts [15,16].

Numerous experimental and theoretical studies have been conducted on alcohol (especially ethanol) dehydration reactions to alkenes over zeolites and other acidic catalysts, reporting a wide variety of reaction barriers under various conditions [14–41]. Activity has been correlated to numerous catalyst properties, including the number of Brønsted acid sites [31] and confinement effects [28], as well as reaction conditions [6]. Note that diethyl ether formation has also been observed during ethanol dehydration over various zeolites [27]; due to the steric hindrances, however, the extension of either of the proposed mechanisms to larger alcohols like isopropanol (IPA) or tert-amyl alcohol (TAA) should significantly reduce the rate of ether formation, for which reason ether formation mechanisms were not considered further in this study.

The preferred concerted dehydration mechanism for ethanol to ethylene over H-ZSM-5 has been established via an exhaustive density functional theory (DFT) study by Kim et al. [22]; however, experimental evidence suggests that the dehydration of secondary and tertiary alcohols may follow different reaction pathways. In the dehydration of tert-butanol over H-ZSM-5, for instance, a tert-butyl cation intermediate has been identified via $^{13}C$ cross polarization magic-angle spinning (CP/MAS) and $^2H$ solid state NMR, providing evidence for an ionic stepwise mechanism (shown in Figure 1 alongside stepwise and concerted mechanisms) [42]. Based on studies with isobutyl alcohol, however, Stepanov et al. proposed a mechanism proceeding via an isobutyl ether intermediate (i.e., a stepwise mechanism) [43,44]. Gayubo et al. [15] reported that iso-alcohols dehydrate more quickly than primary alcohols, though the reason was not discussed. Comparative studies of dehydration activation energies and mechanisms across primary, secondary, and tertiary alcohols are lacking in the literature.

**Figure 1.** Posited (**a**) stepwise, (**b**) concerted, and (**c**) carbocationic mechanisms for tert-amyl alcohol (TAA) dehydration over a zeolite (e.g., H-ZSM-5).

Zeolite choice is also important in conducting such a comparative study because it can affect products formed, reaction mechanisms, and rates due to the variations in electronic structure and sterics. H-ZSM-5 is widely used in the petrochemical industry [15] and is known for its shape-selectivity and high Brønsted acidity adjacent to isomorphous aluminum substitution sites [45]. H-ZSM-5 is a medium pore zeolite with 10 membered straight channels (5.3 × 5.6 Å) and perpendicular sinusoidal channels (5.1 × 5.5 Å). H-BEA zeolite, also widely used in industry, is a large pore zeolite containing three dimensional 12 membered channels, of which two are straight channels (6.6 × 6.7 Å) and one is a sinusoidal channel (5.6 × 5.6 Å) [46]. These structural differences have a demonstrated effect on catalyst function, e.g., reduced adsorption energy of H-BEA for compounds capable of diffusing through both zeolites [47]. The H-AEL zeolite, which is not widely used industrially, contains a single, flatter, elliptical pore (7.0 × 4.1 Å) with a smaller cavity size (maximum diameter of a sphere: 5.64 Å) compared to H-ZSM-5 (6.36 Å) and H-BEA (6.68 Å) [48].

In this study, adsorption of ethanol (EtOH), isopropanol (IPA) and tert-amyl alcohol (TAA) to H-ZSM-5, H-BEA, and H-AEL zeolites as well as reaction mechanisms and kinetics for the dehydration reactions of these chosen alcohols over these zeolites were examined via DFT calculations. Additionally, diffusion of the chosen alcohols in H-ZSM-5 was also investigated using molecular dynamics simulations. Side-by-side comparison of these cases enabled a detailed examination of how confinement impacts on the reaction mechanism and kinetics in a critical reaction class.

## 2. Results and Discussion

### 2.1. Adsorption Energies

Since the adsorption of ethanol on H-ZSM-5 has previously been studied, we first benchmarked our ONIOM (our Own N-layered Integrated molecular Orbital and molecular Mechanics) approach by comparing the calculated adsorption enthalpy in this work with previously published studies, as summarized in Table 1. Note that the previous calculations on the adsorption enthalpies used the periodic DFT-D method and a cluster model. The adsorption enthalpy of ethanol in H-ZSM-5 calculated in this work with 18 T sites incorporated in the QM region was −23.9 kcal mol$^{-1}$, which is in good agreement with Alexopoulos's [49] experimental measurement of −21.3 kcal mol$^{-1}$. Our

result was also consistent with the series of calculations performed by Van der Mynsbrugge [50] using M062X/6-31+G(d) and PBE/6-31+G(d) (with various dispersion corrections) and a large cluster model (46 T sites) as well as periodic functional calculations of H-ZSM-5. They reported values for the adsorption of ethanol on H-ZSM-5 at 400 K ranging from −23 to −34 kcal mol$^{-1}$ [50].

**Table 1.** Adsorption enthalpies (kcal mol$^{-1}$) of ethanol in H-ZSM-5.

| Expt. 400 K [51] | Expt. 300 K [49] | PBE-D pbc, QHA [49] | PBE-D pbc, HA [49] | PBE-D pbc [52] | M06-2X Cluster [50] | This Work, ONIOM |
|---|---|---|---|---|---|---|
| −31.1 | −21.3 | −25.6 | −30.1 | −31.5 | −25.8 | −23.9 |

Alexopoulos et al. [49], Expt. = Experimental results; Van der Mynsbrugge et al. [50], PBE-D = PBE-Dispersion; Lee et al. [51], pbc = periodic boundary conditions; Nguyen et al. [52], QHA = quasi-harmonic approximation; HA = harmonic approximation.

Since the accuracy of the thermal correction for adsorption enthalpy can be influenced by different treatments of the rigid rotor-harmonic oscillator approximation (RRHO) [53], the adsorption energies instead of adsorption enthalpies are reported hereafter in order to compare the ONIOM results with periodic-DFT calculations using VASP. The calculated adsorption energies of ethanol as a function of the size of the QM region using ONIOM (M06-2X/6-311G(d,p):PM6) are shown in Figure 2. Up to the 18 T models, no convergence was observed for any of the three zeolites, possibly because PM6 was used to calculate the interactions between the substrate and the zeolite lower level region in the adsorbed complex, but not in the reference state, i.e., the separated zeolite and substrate. Therefore, as more zeolite atoms are moved from the lower level region into the QM region, the electronic interactions between the newly added QM zeolite atoms and the substrate will be treated more accurately, resulting in a continuous change of the adsorption energy. Due to the long-range nature of these electronic interactions, we speculated that the adsorption energy calculation would not converge quickly, making the adsorption energy calculations using PM6 problematic. However, this should not affect the PES (Potential Energy Surface) calculations using PM6, since PM6 was used to calculate the interactions between substrate and zeolite for both transition state and reactant complex—the error of PM6 should be cancelled out. Hereafter, the QM region was chosen to include a 9 T model for H-AEL, an 8 T model for H-ZSM-5, and a 9 T model for H-BEA, shown in Figure 3.

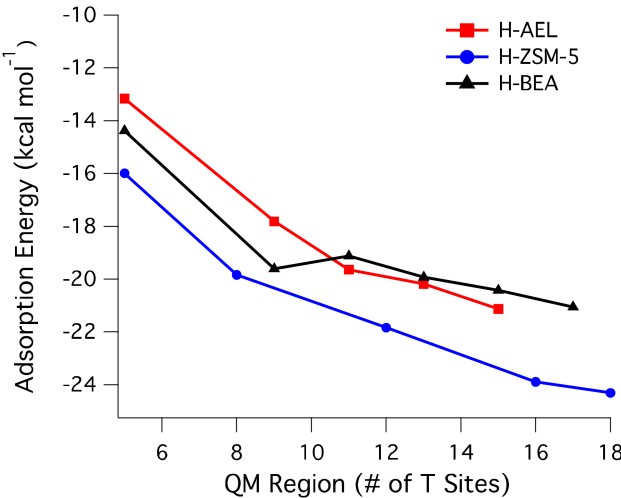

**Figure 2.** Calculated adsorption energies as a function of the size of the QM region using ONIOM (M06-2X/6-311G(d,p):PM6).

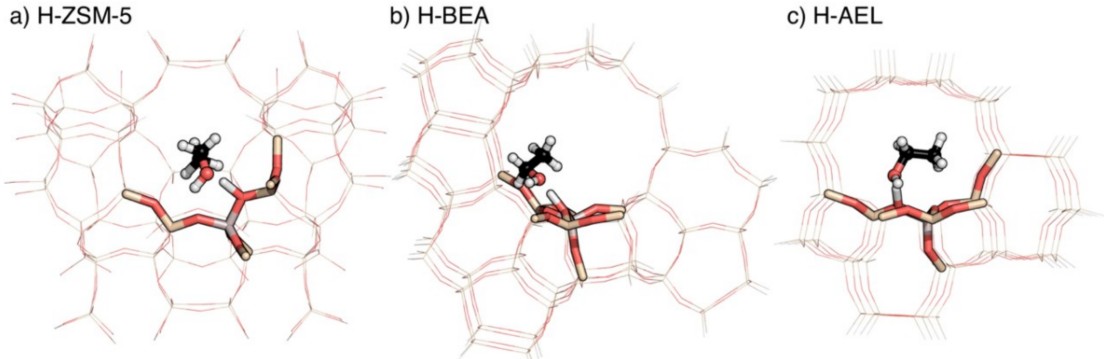

**Figure 3.** Systems studied for ethanol within (**a**) H-ZSM-5, (**b**) H-BEA, and (**c**) H-AEL. The ONIOM high-level DFT (Density Functional Theory) region is shown in "sticks"; the semi-empirical low level is shown in wireframe; and ethanol is shown as "ball and sticks." The zeolite models used for isopropanol (IPA) and tert-amyl alcohol (TAA) were identical; (black: C atom; white: H atom; red: O atom; brown: Si atom; gray: Al atom).

As shown in Table 2, the molecule size of TAA, $5.8 \times 6.7 \times 7.0$ Å, was comparable to the cavity diameter of 6.4 Å in H-ZSM-5 (where cavity diameter was defined as the maximum diameter of a sphere that can be included in the zeolite) [54]. The adsorption strength of TAA in the straight channel of H-ZSM-5 was 11.4 kcal mol$^{-1}$ stronger than in the sinusoidal channel using the 8 T model, indicating that TAA was large enough to be significantly impacted on by steric effects. Hence, the straight channel in H-ZSM-5 was used instead of the sinusoidal channel in all subsequent calculations. Furthermore, this steric effect obstructed the formation of a stable 2-hydrogen-bond (2-HB) binding mode for TAA in the straight channels of H-ZSM-5, as shown in Figure 4.

**Table 2.** Measured molecule size of ethanol, IPA, TAA, and zeolite cavity size. The cavity size of a zeolite was defined as the maximum diameter of a sphere that can be included in the zeolite [54]. A description of the measurement of molecule size is provided in Figure S1.

| Molecule Size (Å) | | Cavity Size (Å) [54] | |
| --- | --- | --- | --- |
| Ethanol | $4.1 \times 4.5 \times 6.5$ | H-AEL | 5.64 |
| IPA | $5.0 \times 5.7 \times 6.1$ | H-ZSM-5 | 6.36 |
| TAA | $5.8 \times 6.7 \times 7.0$ | H-BEA | 6.68 |

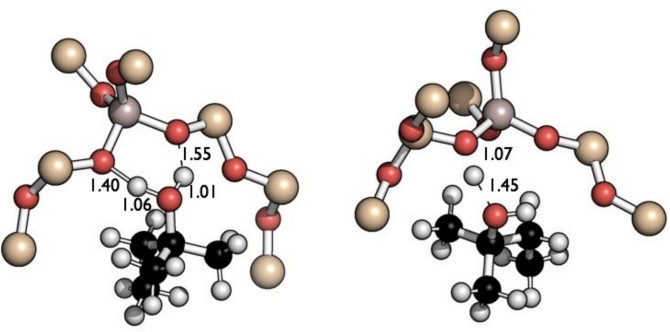

**Figure 4.** Optimized geometries of TAA in H-ZSM-5 with sinusoidal (left) and straight (right) channels; (black: C atom; white: H atom; red: O atom; brown: Si atom; gray: Al atom).

The calculated adsorption energies of EtOH, IPA, and TAA in H-AEL (9 T model), H-ZSM-5 (8 T model), and H-BEA (9 T model) using ONIOM and VASP are shown in Table 3. The adsorption energies calculated for ethanol and IPA were relatively close among the two methods, but they differed significantly for TAA. Moreover, the ONIOM results suggested higher adsorption energies in H-BEA compared to H-ZSM-5 and H-AEL, as well as lesser adsorption energy with increased

branching, while the periodic results demonstrated no clear trend for the substrates in any of the three zeolites. The primary difference in the optimized adsorbed conformations resulting from these two methods were the acidic proton transfers to the substrate in the periodic calculation (as shown in Figure 5), which might be explained by the failure of semi-empirical cluster models to effectively capture long-range electrostatic effects. Indeed, using the Hartree–Fock approximation (capturing exchange and electrostatic interactions without the correlation energy) in place of the semi-empirical PM6 method for the treatment of the lower-level bulk zeolite framework results in geometries of the adsorbed complex similar to those found with periodic boundary conditions in VASP (Figure S2) and adsorption energies that did not vary significantly from one another across the three alcohols in H-ZSM-5 and H-BEA.

**Table 3.** Calculated adsorption energies (kcal mol$^{-1}$) of alcohols in H-AEL, H-ZSM-5, and H-BEA using ONIOM (M06-2X/6-311G(d,p):PM6) and periodic (GGA_PBE and single-point PBE-D2) calculations.

| Substrate | ONIOM | | | VASP | | |
|---|---|---|---|---|---|---|
| | H-AEL | H-ZSM-5 | H-BEA | H-AEL | H-ZSM-5 | H-BEA |
| Ethanol | −17.8 | −19.8 | −19.6 | −18.3 | −21.0 | −22.7 |
| IPA | −16.3 | −16.6 | −18.1 | −18.8 | −14.2 | −23.5 |
| TAA | −6.1 | −9.8 | −16.0 | −15.4 | −17.5 | −20.3 |

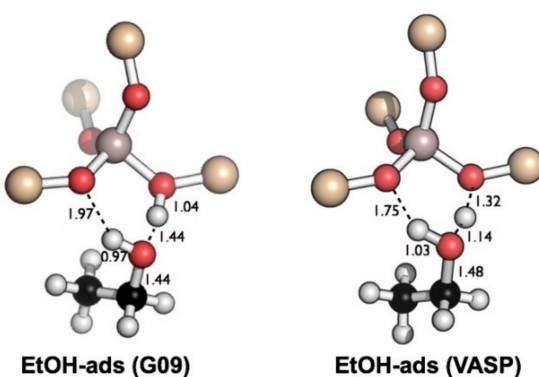

**Figure 5.** Optimized geometries (distances in Å) of adsorbed ethanol in H-BEA using ONIOM (M06-2X/6-311G(d,p):PM6) in Gaussian 09 (G09) and periodic boundary conditions in VASP. (black: C atom; white: H atom; red: O atom; brown: Si atom; gray: Al atom).

Previous studies have shown that increased van der Waal interactions increase adsorption strength until steric effects begin to counter the added stability. Nguyen et al. [52] studied the adsorption of primary C1–C4 alcohols in H-ZSM-5 using a periodic DFT-D method and found that the adsorption strength increased linearly by ~3.5 kcal mol$^{-1}$ per carbon atom due to the increasing van der Waal stability. However, increased branching introduces steric hindrances, and adsorption strength begins to decrease: e.g., butanol decreases in adsorption strength with increased branching: 1-BuOH > 2-BuOH > tert-BuOH by ~3 kcal mol$^{-1}$ for the entire series [55]. The slightly decreasing trend in terms of adsorption strength from ethanol to TAA observed using the ONIOM method presented in Table 3 suggests that the adsorption strength is impacted on more heavily by the degree of branching than the overall number of carbons. However, the periodic DFT-D results in Table 3 show comparable adsorption energies for ethanol, IPA, and TAA in H-AEL and H-BEA, suggesting that the effects of the degree of branching and overall number of carbons on adsorption strength approximately balance each other out; in the intermediate pore case, H-ZSM-5, no trend appears, which could be caused by a greater sensitivity to the exact pore and substrate morphologies when pore and substrate are comparably sized. The discrepancy between the ONIOM and periodic DFT-D results could be due to the inaccurate treatment of the long-range interactions between the free substrate and zeolite in the ONIOM approach.

However, this should not affect the reaction coordinate calculations of adsorbed species and transition states, since PM6 calculated energetics then affect both reactants and products equally.

## 2.2. Reaction Coordinate

Various potential conformations were considered for each structure (i.e., adsorbed reactant, adsorbed product, transition state, or intermediate) along the reaction coordinate. For different conformations with the acidic proton located at the same oxygen, free-energy values for any given structure differed by less than 3 kcal mol$^{-1}$ when using ONIOM (M06-2X/6-311G(d,p):PM6). However, greater variation was observed for free—i.e., not adsorbed—products (depending on orientation within the cavity), consistent with the uncertainty observed in benchmarking reactant adsorption energies using PM6 against Hartree–Fock and periodic VASP calculations.

The energy for ethanol dehydration via a concerted mechanism in H-BEA was 6.6 kcal mol$^{-1}$ smaller than in H-ZSM-5 (Figure 6). Based on an exhaustive transition state search, Kim et al. [22] determined that the ethanol dehydration energy barriers in H-ZSM-5 indicate that the concerted mechanism, proceeding through a late transition state, is preferable over the stepwise pathway. Following such a concerted mechanism, the lower energy barrier for IPA (Figure 7) compared to ethanol in H-ZSM-5 is consistent with what has previously been reported for linear and iso-alcohols [15], suggesting that the stability introduced by increased chain length is more significant than the added sterics introduced by branching.

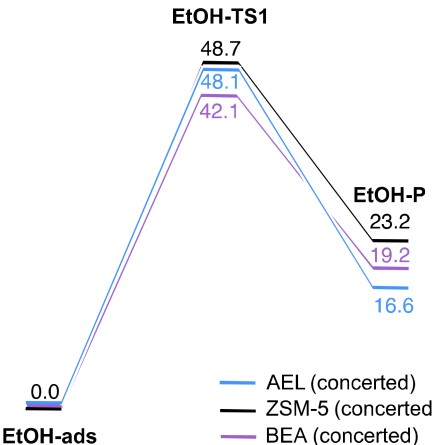

**Figure 6.** Energy values, ΔE (kcal mol$^{-1}$), are reported along the concerted reaction pathway proposed for ethanol (EtOH) dehydration in H-ZSM-5 [22], H-BEA, and H-AEL. All values were calculated using ONIOM (M06-2X/6-311G(d,p):PM6). "EtOH-ads" refers to the ethanol substrate adsorbed at the chosen active site of the respective catalyst. The concerted pathway was consistently a comparable or lower-energy path than the stepwise mechanism [22].

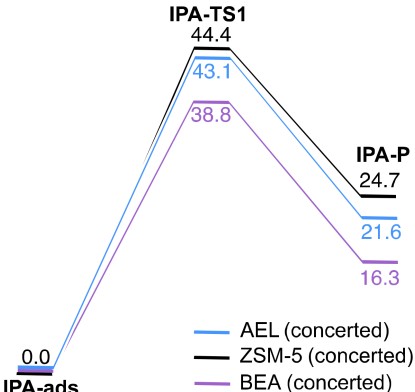

**Figure 7.** Energies ($\Delta E$, kcal mol$^{-1}$) are depicted along the reaction coordinate for the dehydration of IPA, following the concerted mechanism types depicted in Kim et al. [22]. "IPA-ads" refers to IPA adsorbed at the active site of the relevant zeolite.

As shown in Figure S3, H-BEA proceeded through a later transition state than H-ZSM-5, with the proton transferred back to the zeolite 0.28 Å closer to its final Brønsted acid site position than in H-ZSM-5 in the case of ethanol. Given the slightly less stable adsorbed product in H-ZSM-5 relative to H-BEA, the comparable stability of the adsorbed reactants, and similar reaction paths with both catalysts (for ethanol and IPA individually), a smaller intrinsic $\Delta E^{\ddagger}$ in H-BEA than in H-ZSM-5 is consistent with the Hammond–Leffler postulate [56,57]. The H-AEL had a similarly late transition state for both ethanol and IPA (see Figures S3 and S4) but instead had activation barriers comparable to those in H-ZSM-5. This could be due to the strain introduced by the straight channel in H-AEL or the greater significance of long-range electrostatic interactions, which were poorly treated by PM6, in the small H-AEL pores.

For both ethanol and IPA, concerted mechanism transition state intrinsic energies were lower in H-BEA than in H-AEL and H-ZSM-5. As this effect did not vary among the two alcohols, the effects of increased stabilizing electrostatic interactions (due to the increased chain length) and decreased local mobility due to the steric hindrances induced by branching appeared to balance each other out, as was observed for the adsorption energies as well. Note that as the pore size decreased further, steric hindrances began to dominate; in H-AEL, the presence of only one large channel (as opposed to perpendicular, intersecting channels) resulted in substrates being positioned lengthwise within the pore, restricting the number of conformations available to the substrate.

We evaluated the possibility that water molecules may relieve the ring strain in the more constrained, cyclic IPA transition states within the stepwise and concerted reaction schemes. For example, the addition of a water molecule to help shuttle around protons for the second transition state in the stepwise mechanism in H-AEL did expand the ring, but the shape of the channel combined with the presence of two more atoms in the ring resulted in the adsorbed carbon being moved further away from its adsorption site. The C–O adsorption distance increased from an already long 2.6 Å to 2.8 Å, adding to the ionic character of the semi-adsorbed carbon atom and thereby countering the benefit of reduced ring strain. With these two opposing effects, the free energy was approximately unchanged among the two transition states.

Transition states consistent with the concerted and stepwise mechanisms in Figure 1 were not found for TAA. Cyclic transition states were significantly sterically hindered by the addition of a tertiary methyl group and the longer chain length. Previous studies have indicated the stabilization of carbocationic intermediates with increasing carbon number and increased branching [58]. Stepanov et al. [42] also observed such carbocation intermediates during the dehydration of other alcohols. Therefore, we examined a stepwise mechanism with a carbocationic intermediate, as depicted in mechanism "c" of Figure 1. In contrast to the stepwise and alkoxide intermediate schemes, we successfully identified a carbocationic intermediate and associated transition states for the dehydration of TAA

in all three zeolites. Energy values associated with this mechanism for TAA are shown in Figure 8. Ferguson et al. [58] predicted that an ethyl carbocation intermediate does not exist at any relevant temperature, and an isopropyl carbocation only exists in small proportions at high temperatures, e.g., 0.14% at 500 °C; in contrast, a tert-butyl carbocationic intermediate is actually preferred (68.24% at 500 °C) over the alkoxide alternative at all temperatures due to the growing steric effects as the carbon number and the degree of branching are increased. (The carbocation was not nearly as stable for the butyl and sec-butyl carbocations [58]). These conclusions of Ferguson et al. [58] clearly extended to our results for TAA, though the discussions of adsorption energies and of the concerted mechanism above suggest that an increasing carbon number serves to stabilize adsorbates, intermediates, and transition states rather than merely increase steric hindrances.

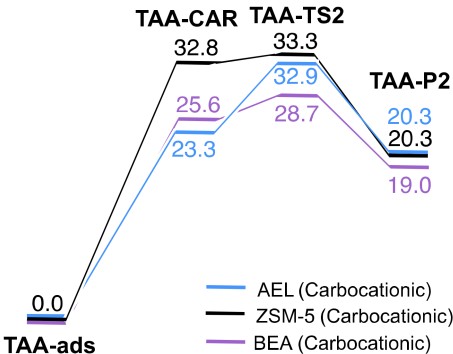

**Figure 8.** Energies ($\Delta E$, kcal mol$^{-1}$) are depicted along the reaction coordinate for the dehydration of TAA through a carbocation intermediate (TAA-CAR) with negligible barrier to 2-methyl-2-butene (TAA-P2). "TAA-ads" refers to TAA adsorbed at the active site of the relevant zeolite.

Note that the thermodynamics of forming 2-methyl–1-butene from TAA (20.7 kcal mol$^{-1}$ in AEL) and 2-methyl-2-butene (20.3 kcal mol$^{-1}$ in AEL) are very similar. Consistent with this, a concerted mechanism transition state could be found for the formation of 2-methyl–1-butene (but not 2-methyl-2-butene) in H-AEL, which is attributed to the slightly decreased steric hindrances; however, the intrinsic barrier was comparable to that found for the carbocationic mechanism (32.9 kcal mol$^{-1}$ for 2-methyl-2-butene and 32.3 kcal mol$^{-1}$ for 2-methyl–1-butene) and 6.6 kcal mol$^{-1}$ lower for 2-methyl-2-butene in H-ZSM-5. Calculations for the carbocationic mechanism indicate that the transition states for the two products were energetically similar, suggesting that sterics of the two products play a smaller role in this mechanism.

Jones et al. [59] studied the confinement of transition states of methanol dehydration in several zeolites with various pore sizes and demonstrated that van der Waals stabilization of transition states is well correlated to the chemical reaction rates, consistent with studies of ethanol over various zeolites [28]. Herein, we assumed confinement influences the transition states and the adsorbed reactant in a similar manner. As shown in Table 3, the calculated adsorption energies of TAA in H-AEL, H-ZSM-5, and H-BEA using periodic-DFT were −15.4, −17.5, and −20.3 kcal/mol, respectively, indicating the stabilization interactions (dispersion and electrostatic interactions) in H-BEA are stronger than in H-AEL and H-ZSM-5. Thus, the lower energy barrier for TAA dehydration observed in H-BEA might be due to the stronger stabilization interactions. Since the product 2-methyl-2-butene exhibits very similar geometry across all three zeolites (as shown in Figure S5), the internal energy will not contribute significantly to the stabilization interactions. As shown in Figure S6, since the transition state of TAA in H-BEA is surrounded by a larger void, the van der Waals stabilization of transition states should contribute less than H-AEL and H-ZSM5. As expected, the VASP calculations suggested that the dispersion corrections to the binding energy of TAA to H-AEL and H-ZSM5 were −28.6 and −22.5 kcal/mol, respectively, while the dispersion energy was only −16.2 kcal/mol in H-BEA zeolite. However, the large void in H-BEA facilitates the formation of a stable 2-HB binding mode of TAA,

which cannot be formed in the smaller pore of H-ZSM-5 due to the steric effects. The more favorable electrostatic interactions overwhelm the dispersion interactions for TAA, resulting in H-BEA exhibiting stronger stabilization than H-ZSM-5.

In AEL, the substrate TAA was substantially surrounded by zeolite atoms along the 1-dimensional channel, as shown in Figure S6, providing a unique opportunity for TAA to form an additional H-bond via a proximate oxygen atom that was not bonded to the Al atom. However, this tight fit may impose severe confinement on the transition state, resulting in alteration of its shape and geometry. As illustrated in Figure S7, the length of the longest principal axis of transition states of TAA increased gradually from 8.2 Å in H-BEA to 8.9 Å in H-ZSM-5 and 10.8 Å in H-AEL while the critical diameter (i.e., the length of the second longest principal axis) decreased from 7.2 Å in H-BEA to 6.7 Å in H-ZSM-5 and 6.4 Å in H-AEL. The pore size decreased in the same order, indicating the transition states were gradually squeezed to accompany the decrease in pore size. The measured sizes of TAA transition states in H-AEL, H-ZSM-5, and H-BEA are summarized in Table 4. The severe confinement from the smaller zeolite pores can also introduce repulsion interactions, resulting in overall less stabilization compared to H-BEA, as supported by the difference among their adsorption energies. Thus, our results indicate that the stabilization effect involves a delicate balance among electrostatic, dispersion, and repulsion interactions between zeolite and substrates.

**Table 4.** Measured size of transition states of TAA and zeolite cavity size.

|         | TAA TS Size (Å)             | Cavity Size (Å) |
| ------- | --------------------------- | --------------- |
| H-AEL   | $5.2 \times 6.4 \times 10.8$ | 5.64            |
| H-ZSM-5 | $5.7 \times 6.7 \times 8.9$  | 6.36            |
| H-BEA   | $5.5 \times 7.2 \times 8.2$  | 6.68            |

*2.3. Molecular Diffusion*

In addition to absorption and kinetic considerations, molecular diffusion can play a significant role in zeolite-catalyzed conversion rates. Indeed, as the molecular size approaches the zeolite pore diameter, one would expect both the transmission function (i.e., the likelihood of the substrate entering the zeolite pore) as well as the diffusivity (indicating the mass transport ability of the substrate within the pore) to be drastically affected. To further complicate matters, standard engineering correlations (e.g., Knudsen diffusion) fail due to the atomic interactions at these scales, vastly overestimating molecular diffusivity by several orders of magnitude [60,61].

Molecular dynamics (MD) simulations of diffusivities of ethanol, IPA, and TAA in H-ZSM-5 were performed to investigate mass transport using the LAMMPS program [62], which has been shown to work effectively for biomass pyrolysis products [63]. A detailed description of the method can be found in the Supplementary Materials. The calculated self-diffusion coefficients in H-ZSM-5 for ethanol, IPA, and TAA were 34, 4, and $0.021 \times 10^{-10}$ m$^2$ s$^{-1}$ at 300 °C, respectively, while the calculated Knudsen diffusion coefficients using a pore diameter of 5.0 Å were 855, 749, $618 \times 10^{-10}$ m$^2$ s$^{-1}$ at 300 °C, respectively. This reveals that Knudsen diffusion at this temperature overpredicts the diffusivity by about an order of magnitude for ethanol and even more for IPA and TAA. While simulations were not run at other temperatures, it is expected that the relative diffusivities will remain approximately the same even though the absolute value can change by an order of magnitude. This assumption is supported by Bu et al. [63], who investigated the change for a series of species (water, methanol, ethanol, glycolaldehyde) 27 °C to 427 °C.

The implication of these diffusivities can best be understood through the Weisz–Prater criterion ($C_{WP}$) parameter, an estimate of the influence of pore diffusion on kinetics in heterogeneous catalytic reactions:

$$C_{WP} = \frac{-r_{EtOH}\rho R^2}{D_e[EtOH]} \tag{1}$$

where $r_{EtOH}$, $\rho$, R, and $D_e$ are the rate of ethanol decomposition, molecule density, molecule radius, and effective diffusivity, respectively. This Weisz–Prater criterion can be used to determine whether the mass transport is limiting the chemical reactions, particularly in the microporous zeolites, where the relative size of the pyrolysis vapors and the zeolite micropores place the systems effectively within the configurational regime. In general, a $C_{WP} > 6$ indicates that the kinetics are certainly controlled by diffusion, while a $C_{WP} < 0.3$ indicates that any mass transfer limitations are negligible. Developing accurate chemical kinetic parameters from experimental studies in the case of diffusion-controlled reactions is challenging, since it requires additional efforts to decouple the intrinsic rate information from the mass transport. Experimental kinetic results in the literature are typically reported to be in the regime where $C_{WP} < 0.3$, but this evaluation is sometimes based on Knudsen diffusion. Phung et al. [28], for instance, ensured that their ethanol kinetic measurements had $C_{WP} < 0.1$; however, they used Knudsen diffusion, which we have shown is at least an order of magnitude smaller than the self-diffusion characterized by MD simulation. While it is conceivable to operate under conditions where the observed kinetics for ethanol or IPA are not mass transport limited, it is unlikely that intrinsic kinetics could be separated from mass transport limitations under such conditions for TAA, which is 2–3 orders of magnitude slower in terms of the calculated self-diffusion coefficients in H-ZSM-5. This has implications for the biomass-derived species that are branched or otherwise sterically hindered.

## 3. Materials and Methods

A unit cell was constructed for each zeolite, with each model containing the zeolite's primary channel(s) at its center (Figure 2). Bonds at the edge of each cell were terminated with hydrosilane groups (Si–H) for H-AEL and H-BEA and hydroxyl groups (Si–O–H) for H-ZSM-5 to be consistent with previous literature [10,64]. In each case, Al was substituted at a single T site. The choice of T site is still a debated topic in the literature and can be dependent on a variety of synthetic and structural parameters [65]. In the current study, the choice of T site was based on thermodynamic arguments and consistent with existing literature. The T12 in H-ZSM-5 was chosen due to the site's preference for aluminum substitution and subsequent protonation [66,67] as well as its maximization of available space for substrate adsorption [50]. The thermodynamically favored site T7 (H-BEA) and site T1 (H-AEL) were chosen for the other zeolites [68]. The H-ZSM-5 model consisted of 319 atoms, incorporating the straight channel (shown in Figure 2a) and the sinusoidal channel perpendicular to it. The BEA model (352 atoms) was larger in order to capture the offset primary channels that were perpendicular to each other (one of which is shown in Figure 2b). The H-AEL model (260 atoms), shown in Figure 2c, only needed to capture its primary channel, as pore sizes in the perpendicular directions were too small (~2 Å) to be relevant. One aluminum atom was embedded in each unit model in place of a silicon atom, and a proton was bonded to an adjacent oxygen atom to balance the charge, thus creating a Brønsted acid site. With only one active site per unit model, the Si/Al ratios in H-ZSM-5, H-BEA, and H-AEL were 75, 105, and 75, respectively. Note this is a reasonable approximation of catalytic fast pyrolysis experiments, where the Si/Al ratio is typically in the range of 50–60.

The ONIOM calculations were performed in Gaussian 09 [69], treating the active site with a high level of theory and the surrounding bulk zeolite with a lower level (Figure 3). The high level was treated with M06-2X/6-311G(d,p) while the low level was treated with PM6 and maintained to the experimentally determined geometry obtained from the *Databases of Zeolite Structures* during optimization [54]. We first investigated the effect of the size of the high-level region on the calculated adsorption energy of an ethanol molecule in H-AEL, H-ZSM-5, and H-BEA (Figure 2). The adsorption energy was calculated by:

$$\Delta E_{ads} = E_{complex} - E_{zeolite} - E_{substrate}$$

where $E_{complex}$, $E_{zeolite}$, and $E_{substrate}$ are the energies of the adsorption complex, the empty zeolite, and the substrate in the gas phase. Various finite-size models varying from 5 T to 18 T models were examined in the benchmark study to identify the optimal size. Subsequently, an 8 T model for H-ZSM-5 and 9 T models for H-BEA and H-AEL were constructed with fixed (rather than periodic)

boundary conditions for further calculations. The substrate and the chosen finite-size portion around the active site (the quantum mechanical (QM) region) were treated with the M06-2X functional, a hybrid functional recommended by Zhao and Truhlar for main group thermochemistry and non-covalent interactions with improved potential energy surface (PES) approximations for isomerization reactions and van der Waal dispersion interactions [70–72]; the 6-311g(d,p) basis set was employed for geometric optimization, frequency, and energy calculations.

The remainder of each model (the bulk of the zeolite framework, named the molecular mechanical, or lower level region) was treated with the semi-empirical PM6 method, reported as having hydrogen-bonding corrections and being accurate in predicting heats of formation of organic molecules (average unsigned error of 4.4 kcal mol$^{-1}$) [73]. The atoms in the lower level region were frozen in place such that while the mobile higher-level region responded to electric effects from the surrounding framework, the framework remained rigid.

Transition states for the posited mechanisms were found with model redundant optimization and frequency calculations, followed by saddle-point optimizations based on the Berny algorithm. The validity of the final transition states was confirmed with frequency and intrinsic reaction coordinate (IRC) calculations.

Previously, we performed an exhaustive search for optimized adsorbed species; the energy differences were small (~5 kcal mol$^{-1}$) in comparison with the reaction barriers [22]. Consequently, we performed only a limited search with a handful of conformations in the present work.

For these adsorption calculations, unless otherwise noted, alcohol molecules were oriented in a 2-HB mode. Consistent with the results of Nguyen et al. [52], ethanol and IPA were placed along the sinusoidal channel in H-ZSM-5, while TAA laid along the straight channel due to the steric hindrances [55]. Both BEA and AEL only have one type of accessible channel (straight), and the molecules were oriented accordingly.

In addition to the ONIOM calculations, periodic DFT calculations were carried out for the adsorption energies of the different alcohol molecules in H-ZSM-5, H-BEA, and H-AEL to confirm adsorption geometries and energies. The aluminum and acidic sites were kept identical with the ONIOM model. The unit cells of H-ZSM-5, H-BEA, and H-AEL contained 290, 194, and 242 atoms, respectively. The periodic calculations were performed using the PBE-D2 method [74,75] with a plane wave basis set implemented in the Vienna Ab initio Simulation Package (VASP, version 5.3.5) [76–79]. The energy cutoff of 400 eV was used, and the Γ-point and a $2 \times 2 \times 1$ k-point mesh were used to sample the Brillouin zones for the gas phase molecules and zeolite systems, respectively. Geometry calculations were carried out using the GGA_PBE methods, and single-point PBE-D2 calculations were carried out on the optimized geometries to account for the dispersion effect. Thermochemical values were analyzed with the *GoodVibes* program [80].

## 4. Conclusions

Catalytic fast pyrolysis is a promising technology that has received much interest in the past decade. However, scale-up of this technology will be hindered given the lack of chemical kinetic models in the open literature. In this study, we investigated three aspects of chemical kinetics—adsorption on the catalytic site, activation energies, and mass transport—as they pertain to the dehydration of ethanol, IPA, and TAA over a series of shape-selective zeolite catalysts.

Adsorption energies are a key component for comparing calculated intrinsic kinetics with experimentally derived apparent kinetics. However, the current study has shown that adsorption energies can be difficult to calculate accurately due to the long-range dispersion forces inherent to zeolites. For ethanol and IPA, the magnitude of adsorption energies was captured relatively well when compared to periodic functional calculations. However, trends with increased substrate branching were inconsistent (Table 3). For TAA, where confinement effects play a larger role, absolute energetics were inconsistent between the methods employed. This is largely true for the broader literature as

well, as depicted for ethanol, in which the calculated absorption energies varied by almost 10 kcal mol$^{-1}$ among methodologies (Table 1).

To better understand confinement effects in zeolite catalysis, we investigated the dehydration of ethanol, IPA, and TAA in H-ZSM-5, H-BEA, and H-AEL via concerted, stepwise, and carbocationic mechanisms. In all three zeolites, DFT calculations revealed that the preferred concerted pathway for ethanol and IPA cannot be found for TAA, where a carbocationic pathway is favored instead. Activation energies were seen to be lower for IPA than ethanol, indicative of the stability added by the increased chain length despite steric hindrances introduced by branching, and even lower for TAA, indicative of the stability of the tertiary carbocation intermediate. Activation barriers were seen to be consistently (albeit slightly) lower in H-BEA compared to H-ZSM-5, which was attributed to a later transition state in the case of the concerted mechanism, while high activation barriers in H-AEL might be attributed to increased strain; a disparity between transition state geometries and energetics suggests that these small pore calculations might be more heavily impacted by poor treatment of long-range electrostatic interactions with PM6.

The ONIOM calculations using PM6 for the MM region were found to converge poorly with the increasing size of the QM region when benchmarked against QM/Hartree–Fock as well as periodic VASP calculations. Future side-by-side comparisons targeting various reaction classes with of a range of substrates and zeolites will reveal the structure–function links necessary to optimize process conditions in zeolite-catalyzed upgrading reactions.

Finally, mass transport can be the most critical parameter when considering experimentally determined kinetics, particularly when zeolites are involved due to the relative size of the substrates and nanopores. Our MD simulations demonstrated that the diffusivity of these three alcohols can vary by over three orders of magnitude, which will significantly complicate endeavors to develop an accurate chemical kinetic model of the catalytic fast pyrolysis process. Any attempt to develop kinetic models will require high-fidelity computational components to effectively decouple intrinsic rate information from mass transport.

**Supplementary Materials:** The following are available online at http://www.mdpi.com/2073-4344/9/9/700/s1, Figure S1: The measured size of ethanol, IPA, and TAA; Figure S2: Optimized geometries of adsorbed ethanol in H-BEA using ONIOM (M06-2X/6-311G(d,p):HF/3-21G) in G09; Figure S3: Optimized geometries of transition states of ethanol in H-AEL, H-ZSM-5, and H-BEA; Figure S4: Optimized geometries of transition states of IPA in H-AEL, H-ZSM-5, and H-BEA; Figure S5: Optimized geometries of products of TAA dehydration in H-AEL, H-ZSM-5, and H-BEA; Figure S6: Transition states of TAA in H-AEL, H-ZSM-5, and H-BEA; Figure S7: The measured size of transition states of TAA in H-AEL, H-ZSM-5, and H-BEA.

**Author Contributions:** Conceptualization, D.J.R. and S.K.; methodology, L.Y.K., L.B., B.C.K., C.L., M.R.N., R.S.A., L.A.C., D.J.R., and S.K.; formal analysis, L.Y.K., L.B., C.L., R.S.A., D.J.R., and S.K.; investigation, L.Y.K., L.B., C.L., R.S.A., D.J.R., and S.K.; visualization, L.Y.K., L.B., B.C.K., and S.K.; writing—original draft preparation, L.Y.K. and S.K.; writing—review and editing, L.Y.K., L.B., B.C.K., C.L., M.R.N., R.S.A., D.J.R., and S.K.

**Funding:** This work was funded as part of the Computational Chemistry and Physics Consortium (CCPC) supported by the US Department of Energy's Bioenergy Technologies Office (DOE-BETO) Contract No. DE-AC36-08GO28308 with the National Renewable Energy Laboratory. C.L. thanks the support from the US Department of Energy (DOE), Office of Basic Energy Sciences, Division of Chemical Sciences, Geosciences, and Biosciences, under Contract DE-AC02-06CH11357.

**Acknowledgments:** Computer time was provided by the Texas Advanced Computing Center under the National Science Foundation Extreme Science and Engineering Discovery Environment Grant MCB-090159 and by the National Renewable Energy Laboratory Computational Sciences Center. The authors would like to thank Robin Cywar for giving us the motivation for the project.

**Conflicts of Interest:** The authors declare no conflict of interest.

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
