# Peer review of "Theoretical Determination of Size Effects in Zeolite-Catalyzed Alcohol Dehydration"

_catalysts, doi:10.3390/catal9090700_

Round 1

Reviewer 1 Report

Sound work. With the limitations correctly identified and described. A good overall view of a chemical process catalysed by zeolites, namely alcohol dehydration. The effect of the zeolite structure is correctly analysed in terms of confinement, although one 1 site was analysed. It is missing a short account on whether the site was chosen for optimum shape/size fitting or if it was just more a random or given (preselected) choice.

The mechanisms have been well described and reviewed and equally well explored and the result rationalised. Also the role of diffusion is adequately included. However the Weisz-Prater criterion is not sufficiently well explained and a few more lines would be helpful since it is not usually employed in the academic literature. The words "too generous" could be refined, and also the "implications other" seems to be a typo instead of, perhaps "implications on the". Diffusivity is not "the speed at which the substrate moves". This needs correction.

Tables 2 and 4 indicate 6.36 cavity size for H-ZSM-5, which I'm not sure about. It could well be the diagonal of the channel intersection, but it needs further clarification since channels are not that large.

The manuscript can be accepted in Catalysts

Author Response

Sound work. With the limitations correctly identified and described. A good overall view of a chemical process catalysed by zeolites, namely alcohol dehydration. The effect of the zeolite structure is correctly analysed in terms of confinement, although one 1 site was analysed. It is missing a short account on whether the site was chosen for optimum shape/size fitting or if it was just more a random or given (preselected) choice.

We thank the reviewer for the comment. As we noted in the Material and Methods section, the choice of T-site was based on thermodynamic arguments and consistent with existing literature.

The mechanisms have been well described and reviewed and equally well explored and the result rationalised. Also the role of diffusion is adequately included. However the Weisz-Prater criterion is not sufficiently well explained and a few more lines would be helpful since it is not usually employed in the academic literature.

We have added a few new sentences in the manuscript to explain the Weisz-Prater criterion:

This Weisz-Prater Criterion can be used to determine whether the mass transport is limiting the chemical reactions, particularly in the microporous zeolites, where the relative size of the pyrolysis vapors and the zeolite micropores places the systems effectively within the configurational regime. In general,CWP> 6 indicates that kinetics are certainly controlled by diffusion while CWP< 0.3 indicates that any mass transfer limitations are negligible. Developing accurate chemical kinetic parameters from experimental studies in the case of diffusion-controlled reactions is challenging since it requires additional efforts to decouple the intrinsic rate information from the mass transport.

The words "too generous" could be refined, and also the "implications other" seems to be a typo instead of, perhaps "implications on the". Diffusivity is not "the speed at which the substrate moves". This needs correction.

We thank the reviewer for the suggestions. We have modified the original sentences accordingly:

Phung et al. [28], for instance, ensured that their ethanol kinetic measurements had CWP< 0.1; however, they used Knudsen diffusion, which we have shown is at least an order of magnitude smaller than the self-diffusion characterized by MD simulation.

This has implications on the biomass-derived species that are branched or otherwise sterically hindered.

Indeed, as the molecular size approaches the zeolite pore diameter, one would expect both the transmission function (i.e., the likelihood of the substrate entering the zeolite pore) as well as the diffusivity (indicating the mass transport ability of the substrate within the pore) to be drastically affected.

Tables 2 and 4 indicate 6.36 cavity size for H-ZSM-5, which I'm not sure about. It could well be the diagonal of the channel intersection, but it needs further clarification since channels are not that large.

We have mentioned the definition of cavity size in the original manuscript.

As shown in Table 2, the molecule size of TAA, 5.8 x 6.7 x 7.0 Å, is comparable to the cavity diameter of 6.4 Å in H-ZSM-5 (where cavity diameter is defined as the maximum diameter of a sphere that can be included in the zeolite) [54].

For clarity purpose, we have added one new sentence on Table 2 in the revised manuscript:

Table 2. Measured molecule size of ethanol, IPA and TAA and zeolite cavity size. The cavity size of a zeolite is defined as the maximum diameter of a sphere that can be included in the zeolite [54]. A description of the measurement on molecule size is provided in Figure S1.

Reviewer 2 Report

In this paper, density functional theory calculations are performed to evaluate the adsorption energies of ethanol, IPA and TAA on zeolites, which are H-AEL, H-ZSM5 and H-BEA. They also carefully investigate the alcohol dehydration in the zeolites and discuss the stabilization factors for reactants, TS and products of the dehydration reactions. I would recommend it for acceptance after the minor points listed below.

line 15: "ONIOM calculations of ... were observed to be inconsistent ... condition calculations. " This sentence is unclear. I don't know what the authors want to say. There is no definition of adsorption (or reaction) energies. The definition should be written in the method section. line 143: what it the value of adsorption strength of TAA in the sinusoidal channel? Readers don't know if there is a significant difference between the adsorption strength of TAA in the straight channel and in the sinusoidal channel. It should be described what atoms are colored balls in all figures showing geometries. Optimized structures other than TS, should be shown in SI. Especially, the geometries of products should be necessary. For example, the sentence starting from "As shown ..." in line 283 is valid when all of the products have a similar structure. If the product structures significantly vary, the stabilization interaction may come from the internal change of molecule geometries. line 291: If they want to discuss the balance between electronic interactions and dispersion interactions, the values of dispersion energy should be written. VASP outputs the dispersion energy. line 383, how is the framework geometry of the lower level determined? If they use the experimental structure, they should write it and refer to the reference. If they use structures optimized by VASP or something, it is also needed how to optimize the geometry.

Author Response

In this paper, density functional theory calculations are performed to evaluate the adsorption energies of ethanol, IPA and TAA on zeolites, which are H-AEL, H-ZSM5 and H-BEA. They also carefully investigate the alcohol dehydration in the zeolites and discuss the stabilization factors for reactants, TS and products of the dehydration reactions. I would recommend it for acceptance after the minor points listed below.

We thank the reviewer for the comments.

line 15: "ONIOM calculations of ... were observed to be inconsistent ... condition calculations. " This sentence is unclear. I don't know what the authors want to say.

The original sentence states “ONIOM calculations of adsorption energies were observed to be inconsistent when benchmarked against QM/Hartree-Fock and periodic boundary condition calculations.” That is to say, the adsorption energies in this work are calculated using three different approaches:  1) ONIOM approach; 2) QM/Hartree-Fock; and 3) periodic-DFT using VASP. The results from ONIOM calculations are different from QM/Hartree-Fock and periodic-DFT calculations. We have offered a possible explanation for this discrepancy in the manuscript.

There is no definition of adsorption (or reaction) energies. The definition should be written in the method section.

We have added one sentence in the Material and Methods section to define the adsorption energy:

The adsorption energy was calculated by

∆Eads= Ecomplex – Ezeolite Esubstrate  ,

where Ecomplex, Ezeolite, and Esubstrateare the energies of the adsorption complex, the empty zeolite, and the substrate in the gas phase.

line 143: what it the value of adsorption strength of TAA in the sinusoidal channel? Readers don't know if there is a significant difference between the adsorption strength of TAA in the straight channel and in the sinusoidal channel.  

As shown in line 145, we have indicated the adsorption strength of TAA in the sinusoidal channel.  “The adsorption strength of TAA in the straight channel of H-ZSM-5 was 11.4 kcal mol-1stronger than in the sinusoidal channel using the 8T model.”

It should be described what atoms are colored balls in all figures showing geometries. Optimized structures other than TS, should be shown in SI. Especially, the geometries of products should be necessary. For example, the sentence starting from "As shown ..." in line 283 is valid when all of the products have a similar structure. If the product structures significantly vary, the stabilization interaction may come from the internal change of molecule geometries.

We have added one sentence in each relevant figure to clarify this:

Figure 3.Systems studied for ethanol within (a) H-ZSM-5, (b) H-BEA, and (c) H-AEL. The ONIOM high level DFT region is shown in “sticks”; the semi-empirical low level is shown in wireframe; and ethanol is shown as “ball and sticks.” The zeolite models used for iso-propanol (IPA) and tert-amyl alcohol (TAA)were identical. (Black: C atom; White: H atom; Red: O atom; Brown: Si atom; Gray: Al atom)

Figure 4.Optimized geometries of TAA in HZSM-5 with sinusoidal (left) and straight (right) channels.(Black: C atom; White: H atom; Red: O atom; Brown: Si atom; Gray: Al atom)

Figure 5.Optimized geometries (distances in Å) of adsorbed ethanol in H-BEA using ONIOM (M06-2X/6-311G(d,p):PM6) in Gaussian 09 (G09) and periodic boundary conditions in VASP.(Black: C atom; White: H atom; Red: O atom; Brown: Si atom; Gray: Al atom)

As the reviewer’s request, we have made one new figure (Figure S5) to illustrate the geometries of the products via TAA dehydration in H-AEL, H-ZSM-5, and H-BEA. As expected, the product exhibits very similar geometry, thus the internal energy will not contribute to the stabilization interactions.

Figure S5. Optimized geometries of products of TAA dehydration in H-AEL, H-ZSM-5 and H-BEA. For clarity purpose, only the products, e.g., 2-methyl-2-butene and water, as well as the Al-O-H atoms are shown in spheres.

Figure S5 (attached in the doc file)

We have also added one new sentence in the revised manuscript to discuss the new Figure S5:

Thus, the lower energy barrier for TAA dehydration observed in H-BEA might be due to the stronger stabilization interactions. Since the product 2-methyl-2-butene exhibits very similar geometry across all three zeolites (as shown in Figure S5), the internal energy will not contribute significantly to the stabilization interactions.As shown in Figure S6, since the transition state of TAA in H-BEA is surrounded by a larger void, the van der Waals stabilization of transition states should contribute less than H-AEL and H-ZSM5.

line 291: If they want to discuss the balance between electronic interactions and dispersion interactions, the values of dispersion energy should be written. VASP outputs the dispersion energy.

As the reviewer requested, we have included the dispersion energies in the discussion.

As shown in Figure S6, since the transition state of TAA in H-BEA is surrounded by a larger void, the van der Waals stabilization of transition states should contribute less than H-AEL and H-ZSM5. As expected, the VASP calculations suggest that the dispersion corrections to the binding energy of TAA to H-AEL and H-ZSM5 are -28.6 and -22.5 kcal/mol, respectively, while the dispersion energy is only -16.2 kcal/mol in H-BEA zeolite.

line 383, how is the framework geometry of the lower level determined? If they use the experimental structure, they should write it and refer to the reference. If they use structures optimized by VASP or something, it is also needed how to optimize the geometry.

We have added one sentence in the Material and Methods section to clarify this:

ONIOM calculations were performed in Gaussian 09 [70], treating the active site with a high level of theory and the surrounding bulk zeolite with a lower level (Figure 3). The high level was treated with M06-2X/6-311G(d,p) while the low level was treated with PM6 and maintained to the experimentally determined geometry obtained from the Databases of Zeolite Structuresduring optimization.
